# Monitoring of Indoor Farming of Lettuce Leaves for 16 Hours Using Electrical Impedance Spectroscopy (EIS) and Double-Shell Model (DSM)

**DOI:** 10.3390/s22249671

**Published:** 2022-12-10

**Authors:** Joseph Christian Nouaze, Jae Hyung Kim, Gye Rok Jeon, Jae Ho Kim

**Affiliations:** 1Department of Electronics Engineering, Pusan National University, Busan 46241, Republic of Korea; 2CAS Corporation, Headquarters, R&D Center, Yangju 11415, Republic of Korea; 3Corporate R&D Center, Hanwool Bio, Yangsan 50561, Republic of Korea; 4Exsolit Research Center, Yangsan 50561, Republic of Korea

**Keywords:** electrical impedance spectroscopy, double-shell electrical model, physiological change, leaf impedance, extracellular fluid (ECF), resistance, capacitance, leaf physiology, crop production

## Abstract

An electrical impedance spectroscopy (EIS) experiment was performed using a double-shell electrical model to investigate the feasibility of detecting physiological changes in lettuce leaves over 16 h. Four lettuce plants were used, and the impedance spectra of the leaves were measured five times per plant every hour at frequencies of 500 Hz and 300 kHz. Estimated *R-C* parameters were computed, and the results show that the lettuce leaves closely fit the double-shell model (DSM). The average resistance ratios of *R*_1_ = 10.66*R*_4_ and *R*_1_ = 3.34*R*_2_ show high resistance in the extracellular fluid (ECF). A rapid increase in resistance (*R*_1_, *R*_2_, and *R*_4_) and a decrease in capacitance (*C*_3_ and *C*_5_) during water uptake were observed. In contrast, a gradual decrease in resistance and an increase in capacitance were observed while the LED light was on. Comparative studies of leaf physiology and electrical value changes support the idea that EIS is a great technique for the early monitoring of plant growth for crop production.

## 1. Introduction

Agriculture is the backbone of human existence on Earth. Research on agriculture has become an important focus. The main goal of farmers is to establish a situation that keeps plants healthy. After changing input parameters such as water, nutrients, light management, the required temperature, etc., farmers have to wait hours to weeks to see the health of their plants improve. The idea is to improve the plant tissue’s metabolic and structural sensitivity before any signs appear. Therefore, the rapid analysis of plant responses is critical to ensuring proper and timely treatment since plant phenology can change quickly. This study aims to provide fast information tracking of physiological changes for the indoor cultivation of lettuce by understanding their relationship with the *R-C* change in the DSM over 16 h. It would also be helpful to know how plant parts, such as fruits, branches, roots, and leaves, work from a physiological point of view.

A variety of techniques are used to estimate plant growth and health. First, image monitoring is an ideal noninvasive and nonspecific detection method. It is used for large-scale field monitoring. Thermal imaging detects heat emitted from an object [1]. RGB imaging uses a digital camera to measure changes in the visual spectrum [2]. Fluorescence imaging often involves a combination of a laser and a camera to excite a fluorescent light [3]. By examining light throughout the electromagnetic spectrum [4], hyperspectral imaging can analyze alterations that are not usually detectable by RGB imaging.

Second, spectroscopic approaches such as X-ray spectroscopy [5], mass spectroscopy [6], and Raman spectroscopy [7] can be modified for use with different samples. RGB imaging does not provide more information than a farmer’s observation, but spectroscopy provides information about chemical elements that help us understand physiology. However, the system is bulky and expensive. Finally, electrical-based sensor approaches use electrical components for in vivo plant monitoring, such as microneedle electrodes [8,9] and organic electrochemical transistor-based sensors [10], to learn about plant physiology. However, they are invasive and destructive.

Evidence shows that EIS is nondestructive, can be used for in vivo monitoring, and provides a faster response. EIS can be implemented at a low cost using vendor-specific chips, such as analog devices [11]. In addition, it is accurate, time-saving, and easy to understand and provides improved tracking information. However, the interpretation is somewhat ambiguous and system-dependent, which may lead to limited functionality. It has been extensively used for the evaluation of plant physiology [12,13], root growth assessment [14], frost hardenability analysis [15], fruit and vegetable quality measurement [16,17,18], food quality measurement [19], and crop production [20]. Most EIS studies today are performed using different test frequencies and 2-, 3-, or 4-electrode configurations to obtain more detailed experimental information.

EIS research has become a useful tool for the investigation of the structural characteristics of plant tissues from an electrical point of view. This opens the possibility for several authors to build equivalent-circuit models for the measured values to simplify and interpret the evaluation of biological tissues [21,22,23,24]. The impedance spectra of plant tissues have been characterized by various models [21,22,23,24]. Zhang and Willison proposed the DSM [24] as an accurate model to study plant tissues with larger vacuoles. The difference between the DSM and previous models [21,22,23] is that it considers the resistance of the vacuole, the capacitance of the vacuolar membrane, and the same compartment that has the previous model. Previous models such as the Hayden model take account of the resistance of the *ECF* and cell wall, the resistance of the cell membrane, the resistance of the cytoplasm, and the capacitance of the cell membrane. The DSM appears to have a circuit part that improves information on plant cell physiology and makes the aforementioned model accurate and easy to use.

Indoor farming offers many opportunities to combine advances in genetics with advances in environmental modification. For example, vertical farming guarantees the quality and quantity of crops regardless of the weather, soil conditions, or impacts of climate change [25]. This enables the opportunity to produce functional or specialized food from staples through environmental control and manipulation [26]. During the life of a plant, cells and tissues undergo physiological changes that are essential for adaptation to their environment. The presence of a rigid cell wall that surrounds the cell membrane is a characteristic that can be used to identify plant cells [27].

The focus of this study is on the change in the *R-C* parameters of the DSM to understand the physiological changes in leaf cell compartments. Leaf cell compartments consist of *ECF*, which includes the cell walls, cytoplasm, cell membranes, vacuoles, and vacuolar membranes. The *R-C* value changes were investigated and compared to the physiological response of leaf cells. A mechanistic understanding of leaf development should include a thorough understanding of water uptake and LED light under certain control factors. It is critical to keep in mind that changes in the physiological state of leaves have an impact on *R-C* values.

This study demonstrates that EIS and the DSM can be used to monitor plants and understand leaf physiological changes. This technique is an in vivo and nondestructive monitoring method. Firstly, a curve-fitting method was used to analyze the inner cell changes in lettuce leaves for 16 h. Secondly, the estimated *R-C* values from the DSM were computed using an optimization algorithm method. Finally, a comparative study was conducted to understand the relationship between the *R-C* changes and leaf physiology. Through this study, a large amount of information on plant physiology can be collected, and its potential can be extended to the early detection of diseases, a lack of nutrients, the effectiveness of supplying nutrients, etc. This could help farmers with crop production by merging productivity, quality, and quantity factors, which will increase the efficiency of crop production. A literature review of previous work is summarized in Table 1.

This research has been organized as follows. Section 2 discusses the theoretical foundations of EIS, the anatomy and structure of plant cells, the DSM, and an LED for indoor plant cultivation. Section 3 proposes an EIS monitoring method for leaf impedance measurements. Section 4 displays the findings of the experiment. Section 5 describes the outcomes of the experiment. Section 6 concludes this study.

## 2. Theoretical Background

### 2.1. Electrical Impedance

Impedance (Z) is a vector consisting of two independent components: a real component or resistance (R) and an imaginary component or capacitive reactance ($X_{C}$). The applied frequency determines how smoothly an electric current flows within biological tissues. Resistance is coupled to resistance pathways through fluids within tissues (*ECF* and *ICF*), and reactance is coupled to capacitive pathways, such as structures in cell membranes. Compared to the real component, the imaginary component often dominates at high frequencies. At that level, the physical barrier to the passage of a current is the internal cell membrane resistance. It only allows an electric current to flow back and forth across the cell membrane [31].

Figure 1 shows the relationship between impedance and its individual components (R and $X_{C}$). It is represented as a vector of a quantity that has both a magnitude (|Z|) and direction, which in this case is represented by the phase angle (θ).

Impedance is an AC characteristic of a circuit that may vary with the operating frequency Z(ω). It is expressed by Equation (1), given below [32]:(1)$$Z\left(\omega \right)=R\left(\omega \right)+jX_{C}\left(\omega \right),$$
where *R* represents the resistance, $X_{C}$ represents the capacitive reactance, and ω is the radian or angular frequency of the applied signal. $j=\sqrt{\left(-1\right)}$ stands for the imaginary number. Therefore, $X_{C}$ and θ can be expressed by Equations (2) and (3), respectively:(2)$$X_{C}=-\frac{1}{\omega C}=-\frac{1}{2\pi fC},\;\;$$
(3)$$\theta =\arctan\frac{X_{C}}{R}$$
where *C* is the capacitance, and the angular frequency is ω=2πf.

### 2.2. Plant Cell Anatomy and Double-Shell Model

Plant leaf cells can exist independently as basic structural and functional units of life [33]. Figure 2 presents the simplified anatomy of a plant cell. The cell wall surrounds the cell membrane, which is made up of two layers of phospholipids and proteins. It is a semi-permeable barrier that lets water pass through freely but blocks its solutes [33]. The cytoplasm is a thick solution that fills each cell and contains dissolved proteins, electrolytes, and glucose. It is moderately conductive. Cell membranes mark the edges of cells, control how molecules enter and exit cells, and have low conductivity. The cell can be considered a conductor surrounded by an insulating envelope containing a substructure with similar properties. Figure 2 shows a cross-section of a single plant cell.

A wide range of electrical models for plant tissue impedance spectra have been presented by numerous researchers [21,22,23,34]. Various electrical models have been compared, and the DSM has been proven to be accurate for biological tissues, especially plant cells with a larger vacuole. The DSM has been validated in several plant investigations, such as impedance measurements conducted on nectarine fruit [17], persimmon fruit [35], kiwifruit [36], and the leaves of *Peperomia obtusifolia* L. and *Brassica oleracea* L. [37]. The results from *P. obtusifolia* L. and *B. oleracea* L. leaves show that resistance tended to be higher at a later stage of leaf development, while capacitance tended to be lower.

Results from those investigations confirm that EIS can provide a large amount of information on plant tissue using the *R-C* values of the DSM. However, none of these studies investigated how *R-C* changes are analyzed to understand the plants’ physiology. Figure 3 shows the DSM proposed by Zhang and Willison [24]. It was found that tissue blocks taken from carrot roots and potato tubers closely fit the data with this model.

When excited by an alternating electric current, tissue shows a frequency-dependent behavior: at low frequencies, cell membranes act as insulating barriers with resistive pathways. Therefore, the size of $X_{C}$ increases when a low-frequency alternating current is provided to the cell, which results in the low-energy current being unable to pass through the cell membrane. A low or no current can pass through the capacitive reactance ($X_{C}$), which corresponds to the cell membrane and the applied signal’s frequency. However, as the frequency of the applied alternating current increases, $X_{C}$ gradually decreases, and the current flows through the cell membrane into the cell. At high frequencies, currents can penetrate cell membranes due to their high capacitance, and a large amount of the current can flow through both the *ICF* and *ECF* (Figure 4). At both low and high frequencies, the path of the conduction current through biological tissues is shown in Figure 4.

### 2.3. LEDs and Indoor Plant Cultivation

Artificial light sources have been used for a long time in the culture and growth room industries. These light sources include TFL, high-pressure sodium (HPS) lamps, metal halide lamps (MHLs), incandescent lamps, etc. TFL has been the most popular in the tissue culture and growth room industries. On the other hand, TFL uses up 65% of the total electricity in a tissue culture lab and is the highest nonlabor cost. As result, these industries are always looking for a better way to use light. LEDs have become a promising light source for plant growth in controlled environments since the development of high-brightness LEDs.

Nhut et al. [38] cultured strawberry plantlets under different blue-to-red LED ratios as well as irradiation levels and compared their growth to fluorescent plant growth. The results suggest that a culture system using LEDs is advantageous for the micropropagation of strawberry plantlets. The study also demonstrates that the LED light source for the in vitro culture of plantlets contributes to the improved growth of plants during acclimatization. Xu et al. [39] used blue LED light (450 nm) on tomato leaves to increase the leaf turgor pressure. Fruit color was reported to be redder, and the yield increased. Han et al. [40] assumed that the ideal light for the growth of lettuce should have an emission spectrum that overlaps or contains the absorption spectrum of lettuce.

LEDs can be selected to target the wavelengths absorbed by plants, which allows manufacturers to customize the wavelengths required to maximize production and limit wavelengths that do not significantly affect plant growth. Therefore, LEDs with variable light quality can be used to increase the value of plants to control morphogenesis and secondary metabolite production more efficiently. A blue LED triggers growth and boosts healthy leaves, while a red LED enhances blooming and fruiting and promotes photosynthesis, which is suitable for lettuce plant growth [40]. Their combination has proven to be an effective lighting source for several crops [41].

## 3. Materials and Methods

### 3.1. Experimental Research Overview

This study aimed to find out how *R-C* changes in a DCM over 16 h are related to changes in the physiology of the plant. Farmers could use this technique to obtain information about the condition of the plant more quickly and apply the proper treatment.

Figure 5 presents a summary of the experiment and analysis that were carried out in this study. An outline of our study is described as follows:▪ Lettuce leaves were measured in vivo using an *LCR* meter for 16 h.▪ The output of the *LCR* meter is the impedance, $Z\left(\omega ,T_{i}\right)$▪ A curve-fitting method was used to estimate the *R* and *C* values of the DSM.▪ Comparative studies between changes in *R-C* values and leaf physiology were performed.

**Figure 5 sensors-22-09671-f005:**
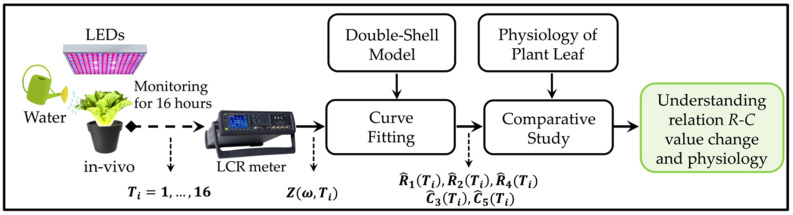
Outline of this work. $T_{i}$ and ω are the measured time index and the angular frequency of measurement, respectively. The impedance $Z\left(\omega ,T_{i}\right)\;$ from the *LCR* meter is time- and frequency-dependent. After curve fitting, the estimated *R* and *C* values (${\hat{R}}_{1}\left(T_{i}\right),\;{\hat{R}}_{2}\left(T_{i}\right)$, ${\hat{R}}_{4}\left(T_{i}\right)$, ${\hat{C}}_{3}\left(T_{i}\right)$, ${\hat{C}}_{5}\left(T_{i}\right)$) were obtained as a function of time with the double-shell model. They were used to understand physiological changes in lettuce.

### 3.2. Indoor Growing Space Setup and Lighting

The experiment was conducted in a growth chamber (temperature: 21.5 ± 1.5 °C; ${\mathrm{CO}}_{2}$ level: 380–400 ppm; relative humidity: 67 ± 0.9%). Four different pots were inserted into the growth chamber. Each pot contained one plant for the study. A LAITEPAKE 4-inch inline fan and a carbon filter were mounted at the top of the chamber to provide airflow inside the chamber. An LED light (GL-225RB-45W, with 165 red 620–630 nm and 60 blue 460–470 nm, Shenzhen city, China) supplied light to the 4 pots arranged at an equal distance from the center of the LED.

The goal of indoor growth is to have the longest, most productive harvest possible, so it is important to provide a good amount of light for the best growth and enough darkness to keep them in their first growth stage. Plants were grown under a photoperiod of 12 h of light and 8 h of darkness. The LED light was installed 50 cm above the leaves. The PPFD output from the light fixtures was measured as 17 $\mathrm{mol}\cdot m^{-2}\cdot {\mathrm{day}}^{-1}$. Plants were watered at 5:30 in the morning and exposed to LED light at 7:30 am. The data considered for this experiment were collected from 5:00 to 20:00 (16 h). Before the experiment, the lettuce plant had to be taken out of the water to observe the physiological changes more easily during measurement. For this, plants were not watered for a few days (approximately 90 h) before the experiment. Dry leaves are associated with more measurement noise compared to leaves containing water and are difficult to connect to the electrode tip.

### 3.3. Impedance Measurement Method and Procedure

Measurements of impedance were taken using a KEYSIGHT E4985AL precision LCR meter. The impedance system used in this study was a user-programmable instrument that provides high reliability and precision. The frequency range used for measurements was from 500 Hz to 300 kHz with an accuracy of 0.05% and good repeatability of measurements at low and high impedances. This instrument can measure both two-terminal and four-terminal test probes. Impedance interfaces use driven shield technology to reduce measurement errors mainly due to input and cable capacitances occurring at high frequencies. The experiment was conducted in a growing chamber for indoor cultivation purposes. Figure 6 shows the measurement methods of the EIS data acquisition system used in this experiment.

A 100 mV generator voltage was used for impedance measurements, and the 201-point list sweep frequency was scanned between 20 Hz and 300 kHz. A four-terminal paired electrode configuration was used for effective and efficient measurement. Measurements were displayed with decimals and units on a high-quality LCD monitor. The electrodes were connected to the lettuce leaves, and the distance between the electrodes was maintained at 3 cm from one tip to another. The *R-C* values were measured with increasing frequency.

The designed experiment process was as follows. Open-circuit calibration and short-circuit calibration were performed before measurements to calibrate the LCR equipment. Using a pair of electrodes adhered to the lettuce leaves, the impedance of the leaves was measured without harming the plants. Throughout all measurement periods, the two electrodes were kept 3 cm apart. After the measurement, the electrodes were washed, dried, and placed on new leaves to prevent unreliable results.The impedance values recorded by the precision LCR equipment were applied to the equivalent circuit, reflecting the growth process of plant tissues. The process of plant growth monitoring was then analyzed and figured out by using a computer to fit the values of the measured impedance parameters to the electrical model. Data were processed using Microsoft Excel 2021 (Microsoft Corporation, Redmond, WA, USA) and graphically analyzed using MATLAB R2022a (The MathWorks, Inc., Natick, MA, USA).

## 4. Results

### 4.1. Frequency-Dependent Resistance and Reactance

To minimize the error in contact impedance measurements from the electrode to the leaf during this experiment, the leaf impedance data were measured every hour five times successively. The greater the number of measurements, the smaller the relative error, and the better the result. A mathematical equation of the mean was used to average the five measurements of impedance [42].

The measured resistance and reactance were defined as ${ℛ}_{p,T_{i}}\left(f,iTry\right)$ and ${ℵ}_{p,T_{i}}\left(f,iTry\right)$, where $p,\;T_{i},\;f,$ and iTry represent the plant index, the measured time index, the measured frequency, and the measurement index, respectively. Equations (4) and (5) showed the mean values of the measured resistance and reactance as follows.
(4)$$mean{ℛ}_{p,T_{i}}\left(f\right)=\frac{1}{N}\underset{iTry}{\sum }{ℛ}_{p,T_{i}}\left(f,iTry\right)\;$$
and
(5)$$mean{ℵ}_{p,T_{i}}\left(f\right)=\frac{1}{N}\underset{iTry}{\sum }{ℵ}_{p,T_{i}}\left(f,iTry\right),\;$$
where N is the number of measurements.

The *R-C* values of impedance for each hour have been plotted. Figure 7 shows how the resistance and reactance of lettuce leaves change with the frequency at 8 a.m. In Figure 7a, the resistance decreases as the frequency increases. As the DSM shows, most of the current flows into the *ECF* at low frequencies. However, at high frequencies, it easily passes through the *ICF*. Figure 7b depicts the reactance of lettuce leaves at the 8 a.m. frequency. A capacitor can be thought of as a variable resistor whose value depends on the frequency being used.

### 4.2. The Proposed Optimization Flowchart

From EIS, leaf impedance was measured, and the obtained data were computed (see Figure 5). The proposed flowchart presented in Figure 8 explains the performed optimization method that was used in this experiment. The algorithm computes the data using MATLAB R2022a App Designer. The flowchart describes the procedure for only one plant at a time, and the process was based on the DSM search parameters. Selected measured data that consist of five measurements at each hour for each resistance and reactance were read, and the mean values of those data were calculated (see Equations (4) and (5)). The search-center points ($R_{1};R_{2};R_{4};C_{3};C_{5}$) and the vector for the step ($\Delta R_{1};\Delta R_{2};\Delta R_{4};\Delta C_{3};\Delta C_{5}$) were initialized. Then, the data were computed, and the estimated impedance value was found using the DSM. An interpolation method helps to predict the search-center-point values ($R_{1}^{k},\;R_{2}^{k},\;R_{4}^{k},\;C_{3}^{k},\;C_{5}^{k})$ in the dataset.

Knowing the measured and estimated impedance, it was possible to determine the absolute impedance error [43]. Equation (6) shows the absolute impedance error ($Z_{err}^{\prime }$)¸ which is the difference between the measured $Z_{m}\left(f,T_{i}\right)$ and estimated $Z_{e}\left(f,T_{i}\right)$ impedance at each measurement hour:(6)$$Z_{err}^{\prime }\left(T_{i}\right)=\underset{f}{\sum }\left|Z_{m}\left(f,T_{i}\right)-Z_{e}\left(f,T_{i}\right)\right|,$$
where $Z_{m}\left(f,T_{i}\right)$ and $Z_{e}\left(f,T_{i}\right)$ are the measured and estimated impedance as a function of frequency and time, respectively.

The fitting algorithm condition to be accepted is that the minimum impedance error must be lower than a threshold ($minZ_{err}^{\prime }\left(T_{i}\right)<Threshold$). To use the fitting algorithm, an optimization method has been used. This process requires efficient and robust techniques. Efficiency is important for this research, which is why this algorithm has been used to solve the unconstrained optimization problem. The equations used below can be found in the book by Edgar et al. [44].

From the starting search-center points ($R_{1};R_{2};R_{4};C_{3};C_{5})$ obtained after interpolation, a search direction was defined s=[−1 0 1], and $minZ_{err}^{\prime }\left(T_{i}\right)$ was minimized in that direction. When the minimum-error impedance condition is not satisfied, new search candidates are set ($R_{1}^{k},\;R_{2}^{k},\;R_{4}^{k},\;C_{3}^{k},\;C_{5}^{k}$), as shown in Equations (7)–(11) below. The search direction is followed by another line of searching. It can be carried out with various degrees of precision. A simple successive doubling of the step size is used as a screening method until reaching an optimal step length value from 0.1 (10%) to 0.001 (0.1%). At this point, the screening search can be terminated, and a sophisticated method was employed to yield a higher degree of accuracy. The step length was set to 0.001 (0.1%) as the minimum step size for the results to be accurate. The search points were updated in a calculation loop, and the estimated curve converged more and more closely to the measured curve until the condition was satisfied ($minZ_{err}^{\prime }\left(T_{i}\right)<Threshold$).

The search-center candidates will move by a step in the search direction defined by s=[−1 0 1] and are defined as $iR_{1},iR_{2},iR_{4},iC_{3},\mathrm{and}\;iC_{5}$ in the $R_{1},R_{2},R_{4},C_{3},$ and $C_{5}$ directions, respectively. At the *k*th stage, the transition from current search-center candidates ($R_{1}^{k},R_{2}^{k},R_{4}^{k},C_{3}^{k},\;C_{5}^{k})$ to new search-center candidates ($R_{1}^{k},\;R_{2}^{k},\;R_{4}^{k},\;C_{3}^{k},\;C_{5}^{k}$) for an iteration of k=k+1 is given by Equations (7)–(11) below.
(7)$$\;\;R_{1}^{k}=R_{1}+\Delta R_{1}=R_{1}+\alpha s=R_{1}+\alpha iR_{1},$$
(8)$$R_{2}^{k}=R_{2}+\Delta R_{2}=R_{2}+\alpha s=R_{2}+\alpha iR_{2},\;$$
(9)$$R_{4}^{k}=R_{4}+\Delta R_{4}=R_{4}+\alpha s=R_{4}+\alpha iR_{4},\;$$
(10)$$C_{3}^{k}=C_{3}+\Delta C_{3}=C_{3}+\alpha s=C_{3}+\alpha iC_{3},\;$$
and
(11)$$C_{5}^{k}=C_{5}+\Delta C_{5}=C_{5}+\alpha s=C_{5}+\alpha iC_{5},\;$$
where α is a positive scalar denoting the distance moved along the search direction, and $\Delta R_{1}=$ a vector from $R_{1}$ to $R_{1}^{k}$ or $\Delta R_{1}\equiv \;\alpha iR_{1}$, which is the vector for the step in the $iR_{1}$ direction (the same applies to all other directions accordingly).

So, $\Delta R_{1}$, $\Delta R_{2}$, $\Delta R_{4}$, $\Delta C_{3}$, and $\Delta C_{5}$ are vectors for the step encompassing both the direction and distance for each center candidate, respectively.

The execution of a unidimensional search involves calculating the value of α and then taking steps in each of the coordinate’s directions, as described in Equations (7)–(11). The repetition of this procedure continues until iteration *n* (*k = n*) is reached. So, the values are minimized. Then, the algorithm terminates when the condition is satisfied ($minZ_{err}^{\prime }\left(T_{i}\right)<Threshold$). The final estimated values (${\hat{R}}_{1}\left(T_{i}\right),{\hat{R}}_{2}\left(T_{i}\right)$,${\hat{R}}_{4}\left(T_{i}\right),{\hat{C}}_{3}\left(T_{i}\right),{\hat{C}}_{5}\left(T_{i}\right)$) were found and stored for all 16 h, and the algorithm process restarted for the following plant.

Figure 9 shows an example of optimization results for one plant measured at one hour, corresponding to plant 1 at 8 a.m. This figure presents a real estimation and an imaginary estimation plotted as a function of frequency. The measurement curve (experiment), fitting curve (curve fit), and error curve have been potted for both real and imaginary graphs.

### 4.3. Physiological Changes in Lettuce Leaves

The results of the optimization algorithm were plotted in groups of each parameter for all four plants as a function of time. These parameters were the cell wall and *ECF* resistance (*R*_1_), the cytoplasmic resistance (*R*_2_), the vacuole resistance (*R*_4_), the cell membrane capacitance (*C*_3_), and the vacuolar membrane capacitance (*C*_5_). Figure 10 shows the average of the internal changes in lettuce leaves monitored and plotted over 16 h. Figure 10a,b show an increase in *R*_1_ and *R*_2_ values after water was supplied and a gradual decrease after light irradiated the leaves. In Figure 10c, there are minor changes in *R*_4_ values. As a result, in Figure 10d,e, *C*_3_ and *C*_5_ show a decrease when water is supplied and a gradual increase after LEDs are turned on.

## 5. Discussion

During the development of a plant, cell growth increases the cell size while cell division increases the cell quantity. This is because of changes in the cell wall, nutrients, and the amount of water the cell can hold. EIS in plant tissues has long been regarded as a technique with application potential in plant physiology [45,46]. However, the results revealed numerous limitations. The ability to access living tissue with minimal injury makes methodological research attractive [24,37]. Many authors have shed light on electrical models used to study biological tissues. Zhang and Willison demonstrated the effectiveness of DSM parameters, which fit plants more closely [24].

Experiments were conducted in vivo, and plants were not watered for a few days to simulate a water shortage. The goal was to observe changes more quickly and clearly. Impedance results were fitted to the model, revealing a low error of approximately 2.1% across the frequency range studied (500 Hz–300 kHz). At hour 5, the plants were dehydrated, and the leaves were wild. Measurements were taken every hour from hour 5 to hour 20 (water was supplied at hour 5:30, and the LED irradiated the leaves at hour 7:30). Figure 10a–e present the final estimates of the resistance ($R_{1},R_{2},$ and $R_{4}$) and capacitance ($C_{3}$ and $C_{5}$) values obtained from the optimization algorithm. These estimation results correspond to the *R-C* parameters of the DSM. From hours 5 to 6, a rapid increase in resistance is observed in Figure 10a,b, while Figure 10d,e present a decrease in capacitance.

The rapid increase in resistance was due to water uptake by the plant. Because the plant was dehydrated, turgor pressure occurred. The turgor pressure was caused by a lack of water, which can be perceived as a physical signal by cell wall sensors, such as ion channels. Increases in the *ECF* and cell wall resistance ($R_{1}$) correspond to a decrease in the stomatal size [47]. A decrease in stomatal conductance occurred before a change in leaf area. This causes the stomata to close, reducing the amount of carbon dioxide available, and due to water stress, the leaves wilt, lowering the metabolic activity of the leaves [47]. The increase in resistance is due to the increased ability of leaf cells to absorb water. In other words, wall extension causes a volume increase, which can be used to accommodate a larger cytoplasmic volume and nucleus or a larger vacuole due to water uptake. Cell expansion is primarily associated with vacuole enlargement [48]. From hour 7 to hour 20 (with the LED on at 7:30), gradual decreases in cell wall resistance ($R_{1}$), presented in Figure 10a, and cytoplasmic resistance ($R_{2}$), presented in Figure 10b, were noticed. This is explained by the cell’s rehydration process [49]. The structures of plant roots, stems, and leaves facilitate the transport of water, nutrients, and photosynthates throughout the plant. The phloem and xylem are the main tissues responsible for this movement. Plants absorb water from their roots and distribute it throughout their compartments. It is assumed that hours 6 to 7 is the time during which the cell absorbed enough water and nutrients. An increase in resistance values and a decrease in capacitance values were observed during water absorption by the leaf cell. In previous studies, the reaction of plant responses to water deficits has been discussed. Plants extensively fold their cell walls in response to water deficits, a process that is quickly reversed during water absorption [50,51,52].

LEDs are known to improve the health of plant leaves. As a result, a healthy cell wall provides mechanical strength to the plant [49]. This may be due to water evaporation and/or photosynthesis in the presence of light and ${\mathrm{CO}}_{2}$ in the air. According to King et al. [53], the decrease occurs because of an increase in free ions or an increase in the cross-sectional area of the cell wall that is permeable to low-frequency current. Other authors [54,55,56,57] back up their findings by agreeing that under growing conditions, water vaporization due to leaf transpiration increased dramatically, while leaf net photosynthesis increased linearly with increasing light.

Growing plant tissues typically consist of 80% to 95% water, according to Slatyer [58]. The vacuole contains approximately 90% of the water, with the remainder distributed between the cytoplasm and apoplast. The vacuolar resistance ($R_{4}$) is presented in Figure 10c. Vacuoles, which are membrane-bound organelles, aid in water balance regulation. There were only minor changes in vacuolar resistance ($R_{4}$) values. The concentration of ions in the vacuole does not appear to change rapidly in response to water and light absorption. This is because the vacuole volume contains approximately 90% of water. According to King et al. [53], the vacuoles have the highest ion concentration, so the change in resistance is negligible. In addition, it was proven that throughout growth, the transfer of ions from the vacuole to the cytoplasm and cell wall had little effect on the vacuole’s absolute concentration [59,60].

Figure 9 presents the total impedance variation in the 500 Hz to 300 kHz frequency range, which includes both real and imaginary components. There is an increase in impedance at low frequencies because an intact cell membrane is like a capacitor of high resistance that envelops intracellular fluids. It was found that the membrane is very resistant at low frequencies of an applied electric field, and only a small amount of the electric current moves through the *ECF* around the cells. This is called high impedance. Because this impedance is mostly capacitive, as the frequency increases, the resistance decreases (Figure 7 and Figure 9) [29].

At high frequencies, since imaginary and real components are small, the membrane impedance approaches zero (Figure 9), and the membrane acts as a short circuit. Then, the electric field passes easily through both the *ECF* and *ICF*. The plant cell membrane acts electrically like a capacitor, with a capacitance that fluctuates with frequency [29]. The presence of dielectric material in the capacitor increases the capacitance value, as suggested by Juansah et al. [61].

Figure 10d,e show that the capacitance value decreased as a result of water uptake. The plant was in poor condition due to the water shortage. The decrease is related to water absorption by the cell. After water absorption and with the addition of the LED, which irradiated the lettuce leaves, an increase in the capacitance value is noticed. It is known that leaf impedance values change with changing cell water content and cell membrane permeability. Therefore, the current flows through the *ECF* at low frequencies, as the components of the layer of the cell membrane capacitance ($C_{3}$) and vacuolar membrane capacitance ($C_{5}$) and organelles (proteins, macromolecules, and other components) have time to polarize. This prevents the flow of an electric current through them, and they act as capacitive components, increasing the value.

In conclusion, the rapid increase observed in resistance and the decrease observed in capacitance after water uptake can be explained as follows: the plant moves from a dehydrated state (low turgor pressure) to a hydrated state (increased turgor pressure), where the process is quickly reversed. It is known that turgor pressure is an ideal means in plant cells through which the energy content of water molecules (water potential) can be adjusted quickly (within a second). It plays a key role in processes such as growth, development, mechanical support, signaling, flowering, and stress responses.

The average resistances were given as $R_{1}=159.07\;M\Omega$, $R_{2}=4.76\;M\Omega ,$ and $R_{4}=1.49\;M\Omega$. The results revealed that resistance $R_{1}$ (*ECF* and cell wall resistance) was higher compared to $R_{2}$ and $R_{4}$ resistances for a ratio of $R_{1}/R_{4}=10.66$ and $R_{1}/R_{2}=3.34$. This result was expected, according to previous results [62], the tiny cross-sectional area, and the low concentrations of mobile ions in plant tissues. Therefore, the average resistance of $R_{4}$ (vacuole resistance) was lower compared to $R_{2}$ (cytoplasmic resistance), with a ratio of $R_{2}/R_{4}=3.22$, which can be explained by the higher ion concentration present in the vacuole compared to the cytoplasm and the cell wall. In plants, the vacuole is bound by a membrane called the tonoplast. The tonoplast facilitates the transport of ions and other materials, making their concentrations higher in the cytoplasm than in the cytoplasm and the cell wall [53,63]. Harker and Maindonald [17] showed that resistance measured at high frequencies in nectarine tissue was mainly related to vacuole resistance. This was expected because vacuoles take up most of the space inside most leaf cells and have higher concentrations of ions.

It was also discovered that there is a dynamic range difference between plant 2 (see Figure 10a,d,e) and the other plants. It was assumed that the slow recovery or growth of plant 2 could be a response to that difference. This possibility was assumed by the authors because of the daily plant observations. When compared to other plants, the plant’s height and leaf growth size differ from those of other plants (unfortunately, this manuscript does not include a table of plant comparisons). To support this premise, more investigation is required.

## 6. Conclusions and Future Work

This study used 16 h of EIS monitoring to examine changes in internal leaf cells in order to assess and understand plant physiology. EIS monitoring is a nondestructive technique with great potential for monitoring the cultivation and development of indoor plants. EIS is sensitive to changes in leaf cells due to water uptake; the responses are quick, and it is simple to use. Furthermore, using the DSM to track the *R-C* change to understand the physiological status of the plant has been shown to be effective and yield good results.

This research highlighted comparative analyses of physiology in relation to a DSM parameter change. Low-frequency resistance measurements reveal significant changes in the cell wall and *ECF*, according to the findings. High-frequency resistance tests, on the other hand, revealed that cells undergo significant changes following water uptake and LED light stimulation. This means that ion movement within lettuce leaves’ intracellular compartments did not change significantly. The ability of a plant variety to limit the growth and development of a specific pest or pathogen and/or the damage they cause when compared to susceptible plant varieties under similar environmental conditions and pest or pathogen pressure is thus understood as resistance from the perspective of plant electrical impedance. So, from the standpoint of plant electrical impedance, resistance can be interpreted as the ability of a plant variety to restrict the growth and development of a specified pest or pathogen and/or the damage they cause when compared to susceptible plant varieties under similar environmental conditions and pest or pathogen pressure.

The DSM application provided a wealth of information that has previously been validated by plant physiology and cell growth studies. This method is most useful for identifying quantitative cellular changes. This technology has the potential to benefit society in the following ways: First, the direct link between *R-C* and plant physiological changes can assist farmers in increasing crop production. Second, EIS, using the DMS parameters of biological tissues, provides information about the plant tissue cellular structure, *ECF*, and *ICF*, which could be extended and used in many other applications by researchers. According to the findings, EIS is a reliable monitoring strategy for nondestructively detecting physiological changes in lettuce leaves in vivo. Because this research viewpoint is new, the potential can be expanded to early disease detection, nutrient deficiency, water quality, water supply, nutrient quality, and so on.

Although the findings of this study are valid for the described experimental conditions, some limitations can be noted. Despite the fact that EIS has many benefits, its system dependence could restrict its functionality. For instance, depending on the system being used, it can be challenging to attach the electrode tips to the leaf, especially for plants with fragile leaves such as lettuce. It can be difficult to precisely interpret the results.

Future research will concentrate on the importance of understanding the relationship between *R-C* parameter changes in the DSM and physiology at night. It is necessary to experiment for several days, such as 72 h. This study has great potential and could be expanded. Additional research is needed, however, in relation to physiological changes as a function of inputs (nutrients, water, etc.), the measurement time range, the time interval, and so on. This could be widely used for the real-time measurement and optimization of growth conditions.

## Figures and Tables

**Figure 1 sensors-22-09671-f001:**
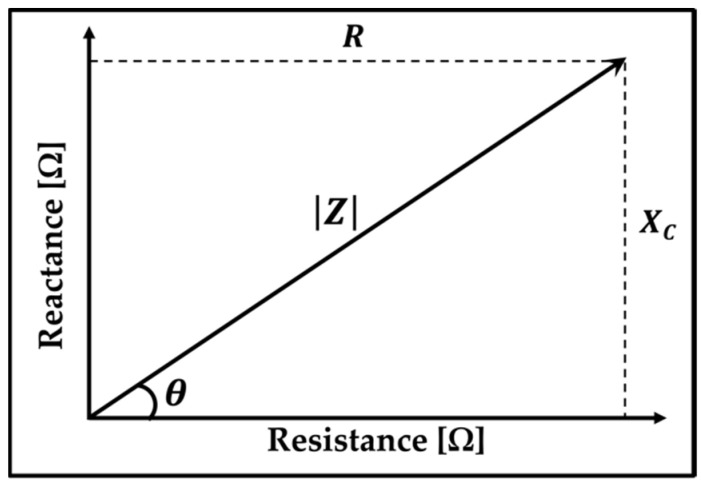
Graph illustrating the impedance vector, where the resistance component’s (*R*) amplitude is shown by a vector on the *x*-axis, and capacitive reactance ($X_{C}$) is on the *y*-axis. A vector from zero to a point that represents both R and $X_{C}$ on the *x*-axis and *y*-axis, respectively, shows the impedance amplitude.

**Figure 2 sensors-22-09671-f002:**
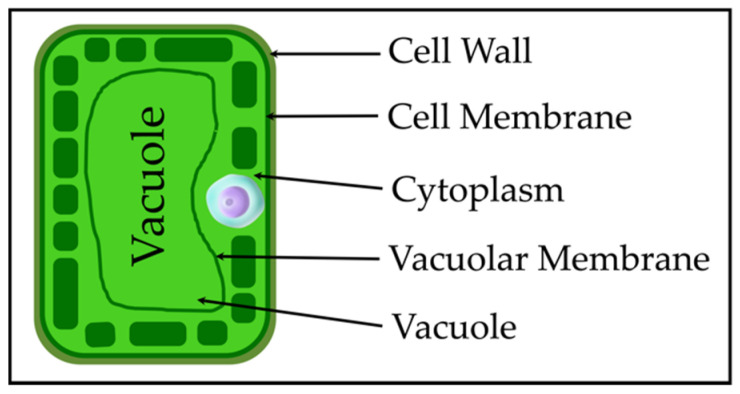
Architecture and constituents of a single plant cell.

**Figure 3 sensors-22-09671-f003:**
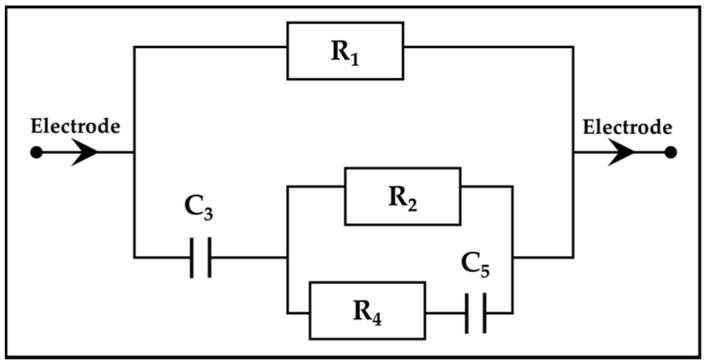
Use of the double-shell electrical model. $R_{1}$ represents the cell wall resistance and the *ECF*, $R_{2}$ represents the cytoplasmic resistance, $C_{3}$ represents the cell membrane capacitance, $R_{4}$ represents the vacuole resistance, and $C_{5}$ represents the vacuolar membrane capacitance.

**Figure 4 sensors-22-09671-f004:**
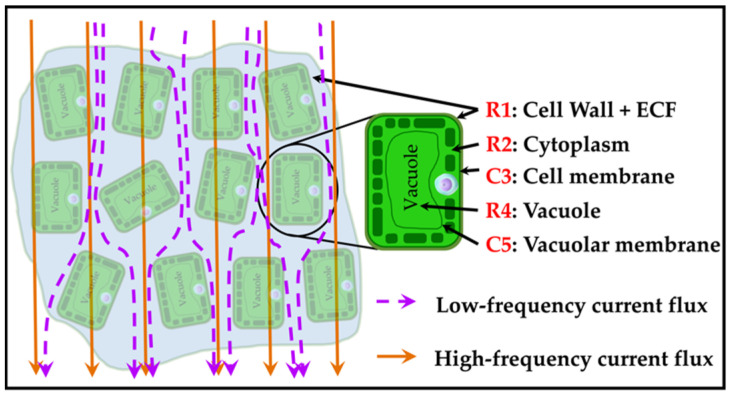
Electric current flows through biological tissue. At low frequencies, current flows through the *ECF*, while at high frequencies, current flows through both the *ECF* and the *ICF*.

**Figure 6 sensors-22-09671-f006:**
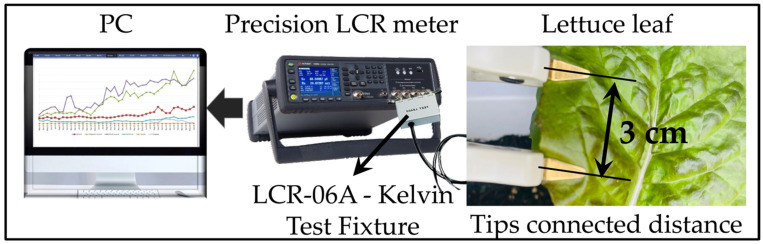
Impedance measurement of lettuce leaves using EIS.

**Figure 7 sensors-22-09671-f007:**
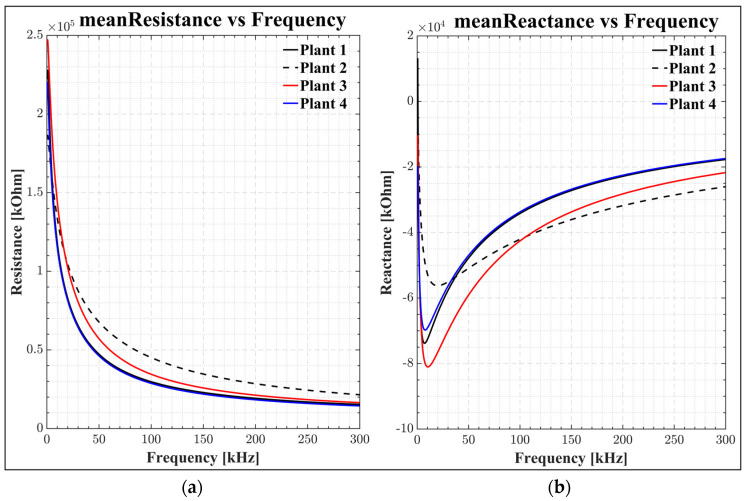
(**a**) Frequency-dependent resistance; (**b**) frequency-dependent reactance graphs, where $mean{ℛ}_{p,T_{i}}\left(f\right)$ is the measured resistance, and $mean{ℵ}_{p,T_{i}}\left(f\right)$ is the measured reactance for various plants (plants 1 to 4) with an average try as a function of frequency.

**Figure 8 sensors-22-09671-f008:**
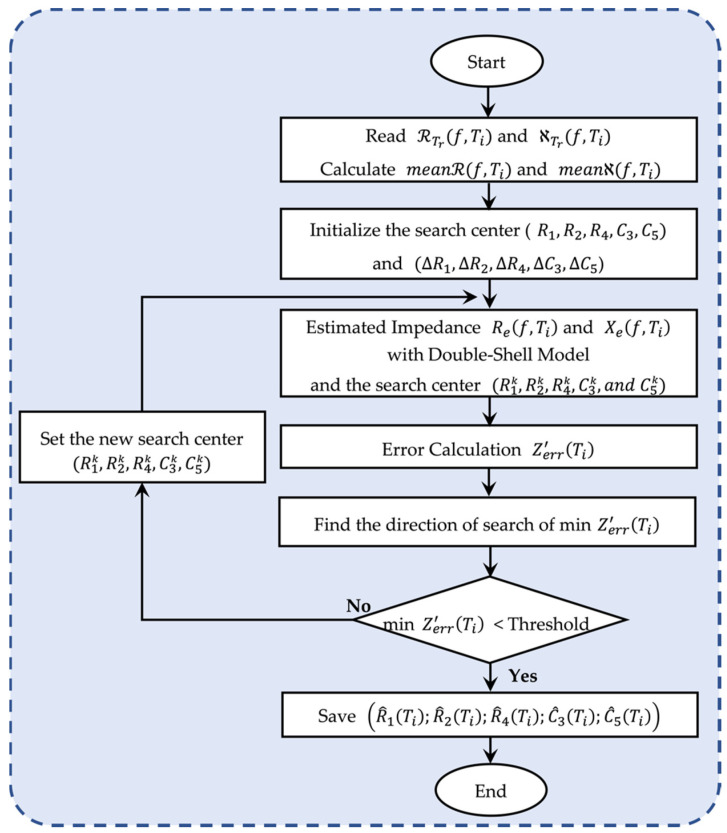
Flowchart of the proposed optimization algorithm to find the *R-C* values. The proposed flowchart is based on mathematical equations and interpolation. The results of the estimated *R* and *C* values (${\hat{R}}_{1}\left(T_{i}\right),{\hat{R}}_{2}\left(T_{i}\right)$, ${\hat{R}}_{4}\left(T_{i}\right)$, ${\hat{C}}_{3}\left(T_{i}\right)$, ${\hat{C}}_{5}\left(T_{i}\right)$) as a function of time were used to understand the physiological changes in lettuce with the DSM. $R_{1}^{k},\;R_{2}^{k},\;R_{4}^{k},\;C_{3}^{k},\;\mathrm{and}\;C_{5}^{k}$ are all search-point candidates. $Z_{err}\left(T_{i}\right)$ is the impedance error as a function of time.

**Figure 9 sensors-22-09671-f009:**
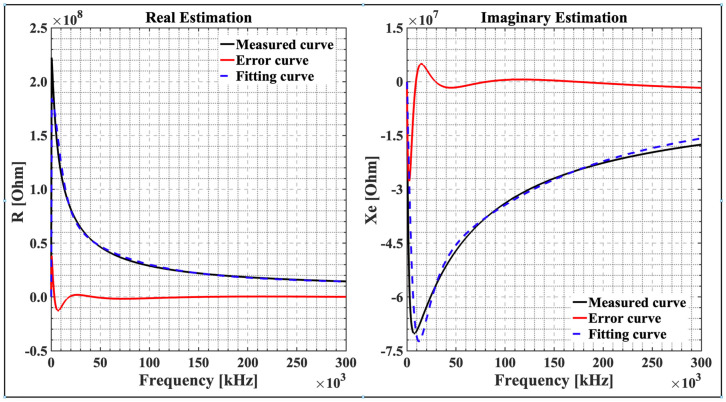
Experimental curve (measurement), curve fit (estimation from optimization algorithm), and error (difference between experiment and curve fit) in real and imaginary estimation graphs. The results were obtained from an optimization program in MATLAB.

**Figure 10 sensors-22-09671-f010:**
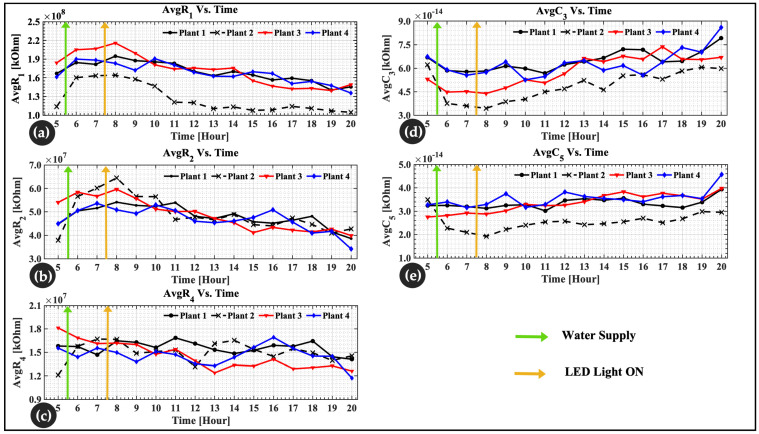
Changes in the inner cell’s physiology over time. (**a**) *ECF* and cell wall resistance; (**b**) cytoplasmic resistance; (**c**) vacuole resistance; (**d**) cell membrane capacitance; (**e**) vacuolar membrane capacitance.

**Table 1 sensors-22-09671-t001:** A brief overview of the literature.

Research Areas	Technique(s)/References	Research Focus	Finding(s)	Application(s)	Limitation(s)	Author(s)
Image monitoring	Thermal imaging [1]	Nondestructive plant physiology monitoring	Stress at an early stage was alleviated. Irreversible damage and yield loss were prevented.	Scales	A few square centimeters can be studied	Chaerle L. et al., 2000.
	RGB imaging [2,28,29,30]	Identification, quantification, and monitoring of plant diseases	Diseases in Cotton, Apple, Grapefruit, and Canadian goldenrod were identified.	Precision in agriculture, plant phenotyping	Low image quality, low detection accuracy	Mahlein, 2016 Camargo and Smith, 2009; Bock et al., 2008; Wijekoon et al., 2008
	Fluorescent imaging [3]	Leaf damage without visible signs, photosynthesis analysis, and fluorimeter comparison of PSM, MFMS, PAM101	A significant difference between same-population leaves and photosynthesis changes was observed. Tree and branch damage patterns were identified.	Natural vegetation, ecological research	Application-dependent	Bolhar-Nordenjampf et al., 1989
	Hyperspectral imaging [4]	Ground-based hyperspectral reflectance of yellow rust disease inoculation. Nutrient-stressed treatment to detect and discriminate yellow rust disease from nutrients.	At major growth stages, four vegetation indices clearly responded to disease. Disease and nutrient stress affected most spectral features. The physiological reflectance index was disease-sensitive.	Disease monitoring and mapping	Cost and complexity	Zhang J et al., 2012
Spectroscopy	X-ray fluorescence [5]	Benchtop XRF to evaluate the elemental distribution change in living plant tissue exposed to X-rays	Higher Zn content than Mn in stems was found. The latter micronutrient presented a higher concentration in leaf veins.	Plant tissue analyses under in vivo conditions	X-rays injure biological tissue	Montanha et al., 2020
	Mass [6]	Monitoring the auxin-regulated nicotine biosynthesis in tobacco and evaluating possible biological effects	Rupture of trichomes and cell damage were observed on spots exposed to Low-Temperature Plasma.	Biosynthesis of plant surface in vivo measurement	Destructive to live cell structure	Martínez-Jarquín et al., 2018
	Raman [7]	High-throughput stress phenotyping of plant measurement	Unique negative correlation between concentration levels of anthocyanins and carotenoids was observed.	Plant stress in vivo	Destructive method	Altangerel et al., 2017
Electrical-based sensor approaches	Microneedle electrodes [8,9]	Measure the xylem sap flow to understand plant physiology	Good adaptation of the microneedle probe in the plant tissue was possible.	Plant physiology (tomato)	Not accurate	Baek et al., 2018; Daskalakis et al., 2018
	Organic electrochemical transistor [10]	Real-time monitoring of the electrolyte of tomato plant’s physiological state	A circadian pattern of variation was revealed, which shows the possibility to detect signs of abiotic stress.	Precision farming, plant physiology	Slow response to plant change, low accuracy	Coppedè et al., 2017
EIS	ZARC-cole and CPE model; CNLS using LEVM7 [14]	Develop EIS for nondestructively evaluating plant root growth of willows	Sum of *R*_1_ and *R*_2_ in the distributed electric model decreased with an increase in root mass.	Root growth assessment	Minor damage due to insertion of needle’s electrode	Repo et al., 2005
	Single-DCE model; CNLS using LEVM v.6 program [15]	Cold acclimation and measurement of frost hardening.	Both quantitative and qualitative changes in cell membranes and water-status were observed. EIS results indicate weaker hardiness than other tests.	Frost hardening capability measurement	System dependent with limited functionality	Väinölä and Repo, 2000
	Different models; nlmin function in S-PLUS [16,17]	Track the electrical change response of fruit physiology and analyze their ripening	Cell wall and vacuole resistance decreased by 60% and 26%, respectively, and membrane capacitance decreased by 9%.	Fruit and vegetable quality measurement	System-dependent, which may limit functionality	Bera et al., 2017; Harker and maindonald, 1994
	Electrical parameter (*Z*, *R*, *Y*, *G*), with simple linear regression analysis [19]	Determining the effect of Total Soluble Solids (TSS) on the electrical conductivity of reconstituted apple juice.	EIS parameters are good for determining TSS content. Rapid determination of the TSS content in different fruit juices, detection of their adulteration	Food quality measurement	Lack of changes in physicochemical qualities	Żywica and Banach, 2015.
	Sensor based on four electrodes and three indexes as indicators of leaf water content [20]	Develop water-saving agriculture and increase water-use efficiency	Negative correlation with all three parameters was observed. Relative water content showed the best correlation with the leaf property.	Crop production (leaf water content)	Existence of a severe fringe effect	Zheng et al., 2014.
	Four different models; CNLS method [24]	Determine the best electrical model for plant tissues analysis	Plant tissue conformed better to a double-shell model than others.	Plant tissue analysis	-	Zhang and Willison, 1991.
	* Finding *R-C* with DSM to understand plant physiology	Monitor and understand leaf physiology for 16 h using double-shell model parameters and a comparative analysis.	$\mathrm{Ratios}\;\mathrm{of}\;R_{1}/R_{4}\;=\;10.66,\;R_{1}/R_{2}=3.34,\;\mathrm{and}\;R_{2}/R_{4}=3.34$ were obtained. The results confirmed previous studies in the literature of. Rapid changes in *R*_1_, *R*_2_, *C*_3_, and *C*_5_ were noticeable after water uptake. Possibility of detecting a plant with slow growth status.	Plant physiology for crop production and precision in agriculture.	System-dependent	Proposed

* The outcome of this study is presented in this table (in the last row) and can be used as a comparison with previous studies.

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
