# Peer review of "Monitoring of Indoor Farming of Lettuce Leaves for 16 Hours Using Electrical Impedance Spectroscopy (EIS) and Double-Shell Model (DSM)"

_sensors, 2022, doi:10.3390/s22249671_

Round 1

Reviewer 1 Report

Reviewer # :  In my opinion, the paper can be published after making minor revisions and some improvements in the presentation of the article, which are as follows :

1.     The abstract did not include any results.

2.     The novelty of the work should also be highlighted.

3.     Equations need references

4.     The quality of the figures are low.

5.     Comparison with previous works are not reported.

Thus, the manuscript should experience the minor revision before acceptance.

Author Response

Dear Reviewer,

Thank you for the valuable comments. All comments were studied point by point and changes were made in the manuscript.

Please see the attachment for the responses raised by your report.

Thank you for your consideration.

Reviewer 2 Report

In this manuscript, the authors detected the physiological changes in lettuce leaves over 16 hours with EIS experiments using a double-shell electrical model. A rapid change in the electrical values was observed after watering the plants. They concluded that EIS is a great technique for early plant growth monitoring. In my view, the manuscript recommend acceptance after major revision.

1.     In comparison, the authors should replace LED with sunlight and strength of the light should be displayed.

2.     I suggest to detect the growth of lettuce leaves for a long time, such as 72 h.

3.     The descriptions of English in the manuscript should be carefully revised and there are numerous errors about grammar and words.

Author Response

Dear Reviewer,

Thank you for the valuable comments. All comments were evaluated point by point and modifications were made to the manuscript. By studying the report form, many criteria have been taken into account, and several sections including Introduction, literature review, Methodology, results, and future study have been updated.

Please see the attachment for the responses raised by your report.

Thank you for your consideration.

Reviewer 3 Report

Article is very interesting. Consider, the following few points to attract the reader's attention and improve the article.

Author Response

(The authors gave the same response as above.)

Round 2

Reviewer 2 Report

The manuscript now can be accepted.